# Neighborhood and Family Characteristics Associated with Adiposity and Physical Activity Engagement among Preschoolers in a Small Rural Community

**DOI:** 10.3390/ijerph192113964

**Published:** 2022-10-27

**Authors:** Emily Hill Guseman, Susan B. Sisson, Jonathon Whipps, Cheryl A. Howe, Madelyn M. Byra, Lucie E. Silver

**Affiliations:** 1Diabetes Institute, Ohio University, Athens, OH 45701, USA; 2Department of Primary Care, Ohio University Heritage College of Osteopathic Medicine, Athens, OH 45701, USA; 3Department of Nutritional Sciences, University of Oklahoma Health Sciences Center, Oklahoma City, OK 73104, USA; 4Department of Natural and Social Sciences, Bowling Green State University Firelands, Huron, OH 44839, USA; 5School of Applied Health Sciences and Wellness, Ohio University, Athens, OH 45701, USA; 6Child Health and Exercise Medicine Program, McMaster University, Hamilton, ON L8S 4L8, Canada; 7Healthy Weight Center, Helen DeVos Children’s Hospital, Grand Rapids, MI 49503, USA

**Keywords:** early childhood, obesity, lifestyle, physical activity, playgrounds

## Abstract

The purpose of this study was to evaluate family and home/neighborhood characteristics associated with physical activity (PA) and adiposity among young children living in a small rural community. Methods: Participants were 30 parents and their youngest child aged 2–5 years. Children wore accelerometers for 7 days. Parents completed questionnaires about family lifestyle behaviors, parenting practices, and home/neighborhood characteristics. Results: None of the family lifestyle behaviors were associated with child BMI percentile. Backyard size was inversely associated with moderate to vigorous physical activity on weekday afternoons (rho = −0.488, *p* = 0.006), as was perception of neighborhood dangers (rho = −0.388, *p* = 0.034). Perceived neighborhood safety (rho = 0.453, *p* = 0.012), the presence of sidewalks (rho = 0.499, *p* = 0.012), and public playground use (rho = 0.406, *p* = 0.026) were each associated with higher weekday afternoon MVPA. Conclusions: Findings suggest neighborhood safety, sidewalks, and use of public playgrounds are positively associated with MVPA among preschoolers, while backyard size and access to play equipment at home are not. These findings have implications for rural communities where space is plentiful but access to community space and sidewalks may be limited.

## 1. Introduction

Trajectories of growth—particularly with respect to body mass index (BMI) and early adiposity rebound—in early childhood are associated with weight status and cardiovascular health in adolescence [1] and adulthood [2,3]. While height gain is fairly stable beginning at about 2 years of age, weight is a phenotype that is more strongly influenced by the environment [4]. As excess weight gain is fundamentally a problem of energy imbalance, research typically focuses on child eating habits and/or physical activity [5]. Additional lifestyle behaviors supported in the literature include engagement in sedentary behaviors—particularly excess screen time—and child sleep habits, where sleep has been shown to be inversely related to childhood obesity [6,7,8]. Despite a growing literature about these lifestyle behaviors, we still do not fully understand the role of parenting practices, such as modeling or shaping PA, in combination with environmental factors.

Although excess weight gain is influenced by a milieu of genetic and behavioral circumstances [9], physical activity is a malleable behavior. Establishing healthy movement patterns in early childhood promotes lifelong physical activity [10]. However, fewer than half of children in the US meet PA recommendations [11] and even fewer meet dietary recommendations with respect to fruit and vegetable intake [12]. Further, preschool-age children are insufficiently active on weekdays and weekends presenting necessary opportunity to enhance movement throughout each day across the week [13,14]. As primary caregivers, parents can promote healthy behaviors by providing support for PA and healthy foods and by modeling desirable energy-balance related behaviors [15]. Parental practices to enhance PA, e.g., overall support, parental positive social control, and general PA encouragement) are associated with children’s habitual PA, but this may vary by child weight status [16,17]. In addition to parenting practices, aspects of the home environment itself can influence PA behaviors in young children [6,18]. However, these studies have not specifically included and targeted families living in rural communities.

Variations in population density and local resources throughout rural regions may influence the availability of recreational spaces and options for childcare, therefore influencing behavior patterns. This is important, as data from two studies examining the relationship between weight status, physical activity levels, and geographical location, suggest that there is a higher prevalence of overweight children in rural areas as compared to urban areas [19,20]. Furthermore, seasonal variation in weather patterns may limit the availability of outdoor recreational activity during much of the year. In geographic regions with more extreme temperatures, parents may restrict children’s physical activity in extremely hot or cold seasons [21]. Few studies have shown relationships between rural and urban environments in children, but in two reviews, data indicated that rural youth were found to be more active in summer, compared to urban youth, who were found to be more active in winter [22,23]. One possible explanation for this difference could be, for instance, lower density of indoor active play options in rural communities in comparison to urban areas.

Due to the unique challenges of living in a rural environment, the approaches to promotion of healthy behaviors in a rural community may need to be different than those in an urban setting [24]. This report seeks to improve our understanding of the links between parent and child behavior patterns in rural environments to help evaluate the need for public health approaches targeted at improving these behaviors in these communities. In this pilot study, we collected data from parent-child dyads to describe the relationships between parenting practices, child lifestyle behaviors, neighborhood and home attributes, and BMI in early childhood among children living in a small rural community with extreme climate patterns.

## 2. Materials and Methods

Participants in this study were recruited from the general population in a small rural community (2010 Census pop. 30,816 [25]) between January 2016–May 2017. Climate patterns in this region, located above 1828 m (6000 ft) in altitude in the Rocky Mountain region, are characterized by long winters, a short growing season, and strong winds [26] Due to budgetary restrictions, we targeted a convenience sample of 30 children between the ages of 2–5 years and their self-identified primary parent (i.e., the parent or guardian primarily responsible for caregiving). Children were excluded if they were unable to walk independently and mothers were excluded if they were pregnant or less than 3 months postpartum. If more than one child in the same family was eligible, the youngest child in the age range was enrolled in the study. Recruitment was conducted via local childcare facilities, preschools, university email list serves, flyers, and posts on social media. Informed consent and verbal child assent were obtained for all participants and the study protocol was approved by the university’s Institutional Review Board (University of Wyoming Protocol #20160107EG01043) and adhered to the principles of the Declaration of Helsinki.

### 2.1. Anthropometry

Parent and child height, weight, and waist circumference were assessed by trained researchers according to standard procedures [4]. Height was assessed in sock feet using a free-standing stadiometer (SchorrProduction, Olney, MD, USA) with the head in the Frankfort plane. Weight was assessed with the participant dressed in light clothing (i.e., shorts and a t-shirt) using an electronic scale (Seca 769; Seca, Chino, CA, USA). Waist circumference was measured at the level of the superior border of the iliac crest using a constant tension fabric measuring tape (Baseline Evaluation Instruments, White Plains, NY, USA). Weight status was determined according to BMI (parents) or BMI percentile (children) [27]. The primary investigator (E.H.G.) fully trained all lab assistants in all anthropometric procedures.

### 2.2. Accelerometry

Children wore the ActiGraph GTX3 (ActiGraph, LCC, Pensacola, FL, USA) on the right ankle for one week (24 h/day) to assess habitual PA and sleep. Ankle-wear was chosen to improve participant compliance and is comparable to data collected using accelerometers worn at the waist (r = 0.77). However, intensity may be overestimated with ankle-wear [28]. Parents completed a wear-time log to record child’s wake time, nap durations, bedtime, and time the accelerometer was not worn. This log was used to validate wear-time, naps, and overnight sleep. Data were included in the analysis if the child wore the accelerometer for at least 4 days, including 3 weekdays and 1 weekend day, for at least 10 waking hours each day [29]. Accelerometer data were analyzed using Puyau ankle cut-points for children [30] using Actilife version 6.13.3 (Actigraph LLC. Pensacola, FL, USA). Child sleep (min/night) was also obtained from the GTX3 data using the Cole–Kripke sleep algorithm and includes only nighttime sleep (naps removed), as not all children napped consistently. Children were required to wear the accelerometer for at least 10 h on at least 4 days for inclusion in the analysis.

### 2.3. Questionnaires

Parents completed the modified preschool PA questionnaire (Pre-PAQ) to provide additional information about family lifestyle behaviors.

Preschool Physical Activity Questionnaire (Pre-PAQ) [31]. Several Pre-PAQ questions address characteristics of the home and environment previously shown to influence child moderate to vigorous physical activity (MVPA); these include a description of backyard size, indication of whether children have access to play equipment (e.g., swing sets) and areas suitable to ride a bike or tricycle, and access to a pool or spa. Questions related to sedentary behaviors include the household number of screen-based entertainment devices, whether there are internet and subscription television services in the home, and an indication of whether the child has a television in their bedroom. Parents also reported the average time spent engaged in screen-based activities, including internet-connected devices, on weekdays and weekend days. Neighborhood questions include presence of open areas, public parks, playgrounds, public swimming pools, gyms offering programs for young children, and clubs offering activities and sports for young children. Finally, parents indicated the degree to which they perceive their neighborhood safe to allow their child to play outside, walk on sidewalks and cross streets, and walk to local businesses.

The Pre-PAQ also includes information regarding attendance at childcare facility centers allowing for calculation of the amount of time children spend in early child education weekly. Parents rated the degree to which their child has an active nature, needs company to be motivated to play, how active the child is compared to other children of the same age, and how often the child eats meals in front of the television. Lastly, the parent used an ordinal scale to indicate how much they agreed with statements regarding encouraging their child to play outside, being physically active with or in front of their child, limiting the child’s activity because of injury fears, focusing on other skills (such as learning letters and numbers), and the degree to which their work schedule limits the time they can spend with their child.

The Pre-PAQ demonstrates good reliability for questions regarding parents’ perception of the neighborhood environment (ICC range 0.96–1.00), child’s activity nature (ICC range 0.87–0.93), use of neighborhoods for activity (Kappa 0.70–0.80), and involvement in organized activities (ICC 0.95), and lower reliability for sedentary activities (ICC 0.44) [31].

### 2.4. Data Analysis

The target sample size for this pilot study (*n* = 30) was limited by available funding; as such, no power calculation was performed. Continuous variables, including subject characteristics, physical activity, screen time, and continuous survey responses, were checked for normality and reported with mean and standard deviation for normally distributed data and with median and inter-quartile range for skewed data. Response frequency counts and percent of sample were computed for ordinal and dichotomous survey responses and for weight status and status with respect to recommendations for PA (60 min/day) and screen time (2 h/day). Due to slightly skewed distributions throughout the dataset, Mann–Whitney U tests were used to assess sex differences and differences according to PA category in subject characteristics, minutes of MVPA, and survey scores (FNPA, CSHQ). After verifying assumptions, Spearman correlations were used to assess the associations between child MVPA and Pre-PAQ responses including backyard size, use of public playgrounds, child personality characteristics, and neighborhood characteristics. All analyses were performed in SPSS v. 26.0 (IBM corp., Armonk, NY, USA) and significance was accepted at two-tailed *p* < 0.05.

## 3. Results

A total of 53 dyads were screened for participation. Two were excluded because they did not meet eligibility criteria (maternal pregnancy) and 21 were unable to be scheduled. Thirty parent-child dyads enrolled in the study (see Table 1 for descriptive characteristics). Child adherence to the accelerometry protocol was excellent and all children met criteria for inclusion in this analysis (mean wear time = 20.6 ± 1.6 h/day). Parents were primarily mothers with a tendency for fathers enrolled in the study (*n* = 6) to be older than mothers (*n* = 24). Approximately 50% of children were girls and child age did not differ between boys and girls. Neither mothers and fathers nor boys and girls differed in terms of BMI (parents), BMI percentile (children), or waist circumference.

As 70% of children (*n* = 21) attended childcare, child MVPA was examined during the afternoon/evening hours (3:00–8:00 p.m.) and separately on weekend days (i.e., times when the child was more likely to be with a parent). Mean child MVPA for the week, on weekday afternoons, and on weekends is shown in Figure 1. Sixty percent of children (*n* = 18) achieved at least 60 min of MVPA daily, and boys and girls did not differ in average total daily or weekend MVPA; however, we found small differences in weekday afternoon PA with boys’ MVPA exceeding girls’ MVPA by about 4.5 min during the 5 h timeframe (median difference 4.55 min, 95% CI −2.00–13.03 min).

Parents generally indicated that they encourage their child to play outside “occasionally” to “all the time” and reported a similar pattern for modeling PA (Table 2). They generally perceived their neighborhoods and parks to be safe (Table 3) and reported that they use parks and playgrounds more frequently than beaches, rivers, and nature reserves (Table 4).

None of the child characteristics differed between those meeting and not meeting PA recommendations. Child MVPA was not related to the degree to which parents encouraged active play outside, whether expressed as weekday afternoon MVPA or weekend MVPA. Minutes of parent-reported screen time (including internet-connected devices) were not associated with MVPA on weekdays or weekend days, but the presence of a television in the child’s bedroom was inversely associated with MVPA on weekends (rho = −0.43, *p* = 0.02).

When examining associations between child characteristics and MVPA, we found that none of the parents’ perception of how active their child is by nature, the child’s need to be motivated to play, or to have company to play were associated with the child’s activity (Table 5). Neighborhood characteristics, however, particularly with reference to neighborhood safety and the presence of sidewalks/footpaths were positively associated with the child’s MVPA, but only on weekday afternoons. Likewise, parents’ perception that there are dangers in local parks were associated with lower child MVPA on weekday afternoons. Finally, we found an inverse relationship between the parents’ description of yard size and child MVPA on weekday afternoons and on weekends. This relationship was somewhat stronger on weekday afternoons (rho = −0.488, *p* = 0.006; 95% CI −0.73, −0.13) than weekends (rho = −0.446, *p* = 0.013, 95% CI −0.70, −0.08) and was unchanged when controlling for parental rating of time limitations or for the child’s participation in organized activities. The strength of the association between backyard size and child MVPA on weekday afternoons was weakened slightly when controlling for public playground use but remained low-moderate in magnitude with a wide 95% confidence interval (rho = −0.372, *p* = 0.047; 95% CI −0.67, 0.02).

## 4. Discussion

The results of this study provide a description of family and home/neighborhood characteristics associated with physical activity (PA) and adiposity among young children in a rural area. No relationship was found between any of the child lifestyle behaviors and BMI percentile. Physical activity participation did not differ between boys and girls when expressed as weekly time spent in MVPA (full day) or in weekend MVPA; however, we did find sex differences equivalent to 1.6 min of MVPA per hour on weekday evenings, which would accumulate a difference of approximately 23 min per week (5 days). Finally, our data suggest an inverse association between yard size and MVPA on weekday evenings and on weekends, such that physical activity was lower among children whose parents reported a larger backyard.

The existence of sex differences in MVPA are well-established in older children and adolescents [32,33,34,35] but these differences are not often seen in younger children. Longitudinal data from the Melbourne InFANT Program Follow-up identified sex differences in MVPA when children were 5 y of age that were not present at the age of 3.5 y or at 19 months [36]. These findings build on a 2016 meta-analysis which determined that, although sex was a determinant of total PA in young children, the difference was limited to light intensity PA rather than MVPA [37]. Another study in Denmark found sex differences in MVPA on weekdays at this young age, and this was limited to the least active children in their sample [38]. In our study, differences in MVPA between boys and girls were limited to the evening hours on weekdays and did not emerge on weekends. This may indicate a difference in the degree to which parents encourage or facilitate PA in the evening hours when time is limited and families have competing demands; unfortunately, our data do not allow us to explore this further. Evidence from Hnatiuk et al. suggests that mothers spend more of their time in the evening modeling or co-participating with their child in sedentary activities than they spend with their child in light PA or MVPA [39], but children in that study were slightly younger (1–3 y) and they did not examine sex differences.

Although parents in our study generally considered their children to be more physically active than other children their age, we did not find a relationship between parent perception of children’s activity and time spent in MVPA measured by accelerometry. This is consistent with studies from the US [40,41] and the UK [42]. Parents may not focus as much on encouraging their children to be physically active if they do not perceive activity levels to be low. Indeed, parents did not accurately assess their children’s PA, as evidenced by the lack of relationship between objectively measured PA and parents’ perception of their children’s activity level. As such, awareness of children’s activity remains an important area for parent education and intervention at the family and community level.

Numerous other personal and neighborhood characteristics may limit young children’s physical activity. These include adult interactions, access to safe facilities that promote physically active play, and affordability of organized activities [43]. Families living in close quarters (e.g., apartment buildings) may limit indoor PA to avoid disturbing others. In cases where the parents do not perceive the outdoor environment to be a safe place to play, this could lead to lower PA among children of any age [43,44,45]. Interestingly, we found that limited backyard play-space was associated with higher physical activity (i.e., an inverse association between backyard size and MVPA). In studies including middle income families, the yard size is not as influential for young children’s PA than the fixed and portable play equipment [46,47]. However, in lower income families, a larger yard was associated with higher PA in preschool children [48]. Parents in our study, however, were generally middle income and perceived their neighborhood to be safe. Thus, the size of the backyard may not have been as influential as those families in lower income areas or with lower perceptions of neighborhood safety. Many parents reported that their children play at public playgrounds frequently, and statistically controlling for use of public playgrounds reduced the relationship between yard size and MVPA slightly. This was specific to public playground use—controlling for use of public parks more generally, or for having play equipment in the home environment, did not change the relationship. These data, in contrast to others’ data [46,47] suggest that simply having access to a play-space and play equipment at home are not enough to encourage MVPA among many young children, but rather that they benefit from access to play structures and other park features. Notably, parents in our sample did report that they encouraged their children to play outside when the weather is suitable frequently (57%) or all of the time (27%), and also reported using parks or open spaces and public playgrounds once per week or more. As such, it does not seem that weather patterns presented a significant barrier to outdoor play for most of these families.

Nearly all of the children in our sample (80%) had at least one sibling in the home; as such, we are not able to determine whether the presence of a sibling influences the relationship between yard size and MVPA. Previous studies have suggested that children with siblings in the home may be more active than children without siblings [49,50,51] but this is not always supported in studies of young children [52]. More research is necessary in exploring birth order and potential sex differences.

Direct or explicit parental modeling of PA is often cited as one approach to increase child PA [53,54]. Researchers have suggested that direct modeling of PA behavior may vary in effectiveness according to child age [55,56,57] and/or that parent and child gender may be a critical modifying factor [58,59]. Despite parents in our sample reporting that they are at least occasionally active “with or in front of” their child, we found no relationship between parent and child PA. As described in a review, several studies indicated parental role modeling and encouragement is associated with PA, but not all studies supported these findings [16]. While this study’s sample size may have been too small to observe a relationship, it may also be due to the varying types of support provided for PA between mothers and fathers and their preschool children [60]. The majority of respondents in our study were mothers, and mother’s co-participation in PA with their young children was low [61]. More information regarding the mode of PA may be critical for understanding these relationships. A parent may put a young child in a stroller and go out for a run—in this case, the parent is directly modeling PA, but the child is inactive. This scenario, and others like it, could be common among parents of young children who need to provide continuous supervision while also prioritizing their own exercise. Additional data are necessary to evaluate the extent to which children who accompany their parent(s) during exercise bouts are physically active later in life.

### Strengths and Limitations

As far as we are aware, this is one of few studies to sample parent-child dyads in a small rural community, particularly in excess of 1500 m in altitude where weather patterns can be more extreme. Despite the rural setting, parents generally reported having easy access to public recreation spaces including parks, open spaces, and playgrounds. Our findings are strengthened by an objective measure of child PA and laboratory assessments of child and parent body size. Further, we conducted an extensive survey evaluation of parent and child lifestyle habits and related parenting practices associated with obesity.

Although the Pre-PAQ questions about ECE attendance are detailed enough to assess time spent in an ECE environment, variability and implausibly high numbers in our data led to concern that some parents may have double-counted ECE time in some instances. As such, we included ECE attendance as a dichotomous variable rather than a continuous one. This may have reduced our ability to detect variations in PA and ST that might be strongly related to time spent in ECE. Additionally, child screen time was reported by parents on the Pre-PAQ and may be subject to response or recall bias.

Parents generally perceived their neighborhood environment to be safe for outdoor play; most households included more than one child, most were two-parent households, and our sample is relatively small. As such, our ability to generalize to other populations is limited. The small sample size in this study resulted in large 95% confidence intervals for many of our analyses and precluded adjustment for possible confounders, reducing certainty. Additional studies with larger samples will be necessary to confirm the findings shown here.

## 5. Conclusions

Our findings highlight the importance of neighborhood characteristics when designing interventions and considering public health approaches to increasing PA in early childhood. This may be especially prescient in low-income rural areas where transportation, indoor options for active play, and access to safe public play-spaces can be limited. Our data suggest that simply having access to plentiful open space is insufficient to encourage PA among young children, and that playgrounds and playmates may be especially important, but additional studies are necessary to clarify the causal mechanisms.

## Figures and Tables

**Figure 1 ijerph-19-13964-f001:**
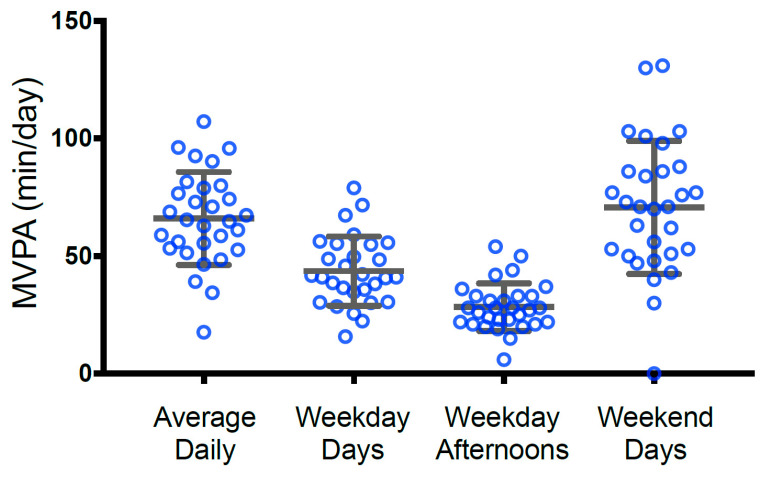
Child physical activity as measured by ankle-worn accelerometer. Values are expressed in min/day and represent the average weekly MVPA, average MVPA on weekday days (except 3:00–8:00 p.m.), average MVPA on weekday afternoons (3:00–8:00 p.m.), and average MVPA on weekend days.

**Table 1 ijerph-19-13964-t001:** Characteristics of the study sample (median and interquartile range except where indicated).

	Median	IQR	Min–Max
Children			
Age (y)	3.3 (mean)	1.0 (SD)	2.2–5.6
Weight (kg)	13.2	3.5	10.6–31.0
Height (cm)	91.2	9.6	82.8–122.6
BMI (kg/m^2^)	15.8	1.6	14.1–20.7
BMI percentile	51.8	56.7	3.3–98.8
Waist circumference (cm)	45.4	4.6	41.2–67.0
MVPA (min/day)	65.1	26.0	17.6–107.2
Afternoon MVPA (min/day)	27.3	11.3	5.5–53.6
Daytime MVPA (min/day)	41.0	22.5	15.8–79.0
Weekend MVPA (min/day)	71.0	35.8	0.0–131.0
Weekday screen time (min/day)	45.0	135	0.0–240.0
Weekend screen time (min/day)	60.0	90	0.0–315.0
Parents			
Age (y)	38.0 (mean)	7.8 (SD)	26.4–49.5
Weight (kg)	73.2	27.5	51.8–124.5
Height (cm)	166.4	7.9	152.7–192.7
BMI (kg/m^2^)	26.6	9.3	15.8–43.9
Waist circumference (cm)	88.2	23.4	65.0–118.9
MVPA (min/day)	69.0	99.8	5.0–277.0

**Table 2 ijerph-19-13964-t002:** Frequency of parent responses to Pre-PAQ questions about PA encouragement and modeling and the extent to which their work limits time with their child.

	Never	Rarely	Occasionally	Frequently	All the Time
Encourage to play outside	0 (0%)	0 (0%)	5 (17%)	17 (57%)	8 (27%)
PA with or in front of child	1 (3%)	1 (3%)	8 (27%)	15 (50%)	5 (17%)
Work limits time of child	3 (10%)	5 (17%)	13 (43%)	8 (27%)	1 (3%)

**Table 3 ijerph-19-13964-t003:** Frequency of parent responses to Pre-PAQ questions about neighborhood and park safety.

	Strongly Disagree	Disagree	Agree	Strongly Agree
Safe to play outside in neighborhood	0 (0%)	2 (7%)	10 (33%)	18 (60%)
Crime makes it unsafe to walk in neighborhood	23 (77%)	7 (23%)	0 (0%)	0 (0%)
Dangers in local parks	16 (53%)	11 (37%)	1 (3%)	2 (7%)

**Table 4 ijerph-19-13964-t004:** Frequency of parent responses to Pre-PAQ questions about open space and public playground use.

	Rarely	Once a Month	A Few Times a Month	Once a Week	A Few Times a Week	Daily
Uses beaches, rivers, nature reserves	10 (33%)	7 (23%)	3 (10%)	4 (13%)	5 (17%)	1 (3%)
Uses parks or open spaces	1 (3%)	3 (10%)	3 (10%)	9 (30%)	10(33%)	4 (13%)
Uses public playgrounds	1 (3%)	3 (10%)	2 (7%)	9 (30%)	12 (40%)	3 (10%)

**Table 5 ijerph-19-13964-t005:** Spearman correlations between child, neighborhood, and home characteristics with total child MVPA (*r*) and child MVPA on weekday afternoons (*r’*).

	*r*	*p*	95% CI	*r’*	*p’*	95% CI’
My work schedule or other commitments limit the time I have to PLAY with my child (never = 1, all the time = 5)	−0.031	0.871	−0.39, 0.33	0.090	0.635	−0.28, 0.44
It is safe for my child to play outside in our neighborhood (strongly disagree = 1, strongly agree = 4)	0.330	0.075	−0.04, 0.62	0.453	0.012	0.09–0.71
There are useable sidewalks or footpaths on most of the streets located in my area (strongly disagree = 1, strongly agree = 4)	0.293	0.116	−0.08, 0.60	0.499	0.012	0.15–0.74
There are major barriers or dangers to walking with my child in my neighborhood that make it difficult to get from place to place (strongly disagree = 1, strongly agree = 4)	−0.232	0.216	−055, 0.14	−0.339	0.067	−0.63, 0.03
There are dangers in the local parks so I avoid taking my child there (strongly disagree = 1, strongly agree = 4)	−0.052	0.787	−0.40, 0.31	−0.388	0.034	−0.66, −0.02
Backyard description (no yard at all, no private yard, small yard, medium yard (standard block of land, <¼ acre), large yard (¼ acre or more)	−0.338	0.068	−0.63, 0.04	−0.488	0.006	−0.73, −0.13
Access to play equipment at home (including swing set, climbing gym, slide, etc.)	−0.102	0.591	−0.45–0.27	0.138	0.468	−0.24, 0.48
My child has a very active nature (never = 1, all the time = 5)	0.351	0.057	−0.02.0.64	0.347	0.061	−0.03, 0.64
My child needs me to motivate him/her to play (never = 1, all the time = 5)	−0.184	0.329	−0.51, 0.19	−0.221	0.240	−0.54, 0.16
My child needs company to be motivated to play (never = 1, all the time = 5)	−0.158	0.405	−0.49, 0.22	−0.242	0.198	−0.56, 0.13
How often does your child use parks or open space to play and be physically active in a typical month when the weather is suitable? (rarely = 1, daily = 6)	0.079	0.676	−0.43, 0.29	0.230	0.221	−0.15, 0.55
How often does your child use public playgrounds to play and be physically active in a typical month when the weather is suitable (rarely = 1, daily = 6)	0.176	0.351	−0.20, 0.51	0.406	0.026	0.04–0.68

## Data Availability

Study data are available from the corresponding author upon reasonable request.

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
