# Peer review of "Neighborhood and Family Characteristics Associated with Adiposity and Physical Activity Engagement among Preschoolers in a Small Rural Community"

_ijerph, 2022, doi:10.3390/ijerph192113964_

Round 1

Reviewer 1 Report (Previous Reviewer 2)

I appreciate the effort made by the authors to modify some of the comments received. However, there are other methodological problems that have not been solved or have not been solved correctly. Indeed, as  I said the questionnaires used by the authors are validated in English, however, it has not been indicated whether these questionnaires were validated in the language of the participants. In addition, the number of participants was very small, which makes me question whether the results are really the ones presented. Therefore, I do not consider this study to be of sufficient quality to be published in this journal. I hope you understand my point of view. 

Author Response

We thank the reviewer for their time. Responses are below:

Indeed, as  I said the questionnaires used by the authors are validated in English, however, it has not been indicated whether these questionnaires were validated in the language of the participants.

Author response: The participants in this study were all English-speaking so the issue of survey validation in the language of the participants does not apply.

In addition, the number of participants was very small, which makes me question whether the results are really the ones presented.

Author response: It is true that the sample is small and that this could lead to spurious findings. However, our data and statistical analyses are sound. We have addressed this in the limitations of the manuscript. It is not possible to increase the sample size at this time.

This manuscript is a resubmission of an earlier submission. The following is a list of the peer review reports and author responses from that submission.

Round 1

Reviewer 1 Report

accept

Reviewer 2 Report

Comments to the Author:

I thank to the editors for the opportunity to review this study, beside I would also like to congratulate the authors for the made effort in their study. The present manuscript by Guseman et al., analyzed “Neighborhood and family characteristics associated with adiposity and physical activity engagement among preschoolers in a small rural community.”, the authors aim of the present study was to collected data from parent-child dyads to describe the relationships between parenting practices, child lifestyle behaviors, neighborhood and home attributes, and BMI in early childhood among children living in a small rural community with extreme climate patterns. The main problem with this study is the number of subjects, which, despite being a pilot study, does not have enough subjects to obtain a statistical power to provide conclusive data. In addition, there are a multitude of methodological mistakes that should be taken into consideration. Authors

1.     Numerous studies show that the rural population is more physically active than the urban population. In fact, the type of work performed in rural life is much more physically demanding than work performed in the city, whatever the environmental conditions given that the work must be done. When we talk about the school environment, the same thing happens, rural school children have a higher physical activity than city children, several articles show these results.  In this way, we want to express our opposition to the hypothesis shown by the authors in their study.

2.     First of all, the authors make an introduction focused on the general rural population, however the authors have developed a study with a very special population given that they were located at 1820 meters of altitude. Therefore, the introduction should focus on studies conducted in this type of environmental situation.

3.     Second, authors should indicate the name of the institution that provides the ethics committee. In addition, the authors have not indicated in the methods section whether their study was based on the principles of the Declaration of Helsinki.

4.     I miss a description of how the experimental design was carried out. When the children first arrived to be measured, was it in the afternoon or in the morning, did they come in groups of 5 or all at the same time. In other words, a schedule of how the data were collected is needed in the manuscript.  

5.     On the other hand, and very important, I note that the questionnaires that were used are in English, but the authors of the study worked with Chinese-speaking participants, or am I mistaken? Therefore, the following question arises, is there a Chinese version of the questionnaires that were used in your study? If so, cite it, if not, we have a problem because the questionnaire must first be validated in the language you want to use, and once validated it can be used.  

6.     However, the biggest problem is the number of participants, since only 30 children participated. How was the number of subjects calculated, to obtain sufficient statistical power to make the results valid and reliable? From my point of view the number of subjects is very small despite being a very special population. I think it would have been necessary to have at least 300 participants to take into consideration the results obtained. For the next study I recommend that you use the G-Power program, it will be very helpful and will give you a sufficient number of participants for your study.  

7.     The first paragraph of the discussion should be much better structured. First it should be the main aim of the study and then the most relevant results. It is not necessary to add more information because it confuses the reader.

8.     The authors should be clearer about the strengths and limitations of their study.

Minor Comments.

-       This sentence “weight is a phenotype that is more strongly influenced by the environment” must be accompanied by at least one reference.

-       This sentence “Furthermore, seasonal variation in weather patterns may limit the availability of outdoor recreational activity during much of the year” must be accompanied by at least one reference.